# Identification of Quantitative Trait Loci Controlling Root Morphological Traits in an Interspecific Soybean Population Using 2D Imagery Data

**DOI:** 10.3390/ijms25094687

**Published:** 2024-04-25

**Authors:** Mohammad Shafiqul Islam, Amit Ghimire, Liny Lay, Waleed Khan, Jeong-Dong Lee, Qijian Song, Hyun Jo, Yoonha Kim

**Affiliations:** 1Department of Applied Biosciences, Kyungpook National University, Daegu 41566, Republic of Korea; shafik.hort@gmail.com (M.S.I.); ghimireamit2009@gmail.com (A.G.); layliny22@gmail.com (L.L.); waleedkhan.my@gmail.com (W.K.); jdlee@knu.ac.kr (J.-D.L.); johyun@knu.ac.kr (H.J.); 2Department of Integrative Biology, Kyungpook National University, Daegu 41566, Republic of Korea; 3Department of Agriculture, Noakhali Science and Technology University, Noakhali 3814, Bangladesh; 4Soybean Genomics and Improvement Laboratory, USDA-ARS, Beltsville Agricultural Research Center, Beltsville, MD 20705, USA; qijian.song@usda.gov; 5Upland Field Machinery Research Center, Kyungpook National University, Daegu 41566, Republic of Korea

**Keywords:** soybean, root morphology, QTL mapping, average diameter (AD), root volume (RV), link average diameter (LAD), SNP, candidate genes

## Abstract

Roots are the hidden and most important part of plants. They serve as stabilizers and channels for uptaking water and nutrients and play a crucial role in the growth and development of plants. Here, two-dimensional image data were used to identify quantitative trait loci (QTL) controlling root traits in an interspecific mapping population derived from a cross between wild soybean ‘PI366121’ and cultivar ‘Williams 82’. A total of 2830 single-nucleotide polymorphisms were used for genotyping, constructing genetic linkage maps, and analyzing QTLs. Forty-two QTLs were identified on twelve chromosomes, twelve of which were identified as major QTLs, with a phenotypic variation range of 36.12% to 39.11% and a logarithm of odds value range of 12.01 to 17.35. Two significant QTL regions for the average diameter, root volume, and link average diameter root traits were detected on chromosomes 3 and 13, and both wild and cultivated soybeans contributed positive alleles. Six candidate genes, *Glyma.03G027500* (transketolase/glycoaldehyde transferase), *Glyma.03G014500* (dehydrogenases), *Glyma.13G341500* (leucine-rich repeat receptor-like protein kinase), *Glyma.13G341400* (AGC kinase family protein), *Glyma.13G331900* (60S ribosomal protein), and *Glyma.13G333100* (aquaporin transporter) showed higher expression in root tissues based on publicly available transcriptome data. These results will help breeders improve soybean genetic components and enhance soybean root morphological traits using desirable alleles from wild soybeans.

## 1. Introduction

Worldwide, soybean [*Glycine max* (L.) Merr.] is a major economic, legume, and oil crop that is essential for food, feed, fodder, and industrial production [1,2,3]. The main soybean-producing countries are the United States of America, Brazil, and Argentina, accounting for 33%, 29%, and 19% of the total global production, respectively (www.soystats.com, accessed on 20 April 2023). It is considered that cultivable soybeans evolved from wild soybeans (*Glycine soja* Seib.et Zucc) approximately 3000–6000 years ago in China [4,5,6].

Roots are the most important features of a plant for transporting and capturing water, supplying nutrients, and regulating floods [7,8]. Root traits, including root length, diameter, and surface area, act as a harbor for plants, enhancing plant growth by taking up or transporting water and nutrients and regulating nutrient availability through symbiotic association with microbes [9,10]. Root morphology helps regulate plant efficiency under different environmental conditions and plays a vital role in recovering drought avoidance by increasing water absorption from the soil layers [11,12]. Root morphological traits also play a significant role in minimizing the different types of stress. Hence, soybean root growth significantly affects soybean production and adaptation to morphological, physiological, and anatomical changes under stress conditions [13]. It is widely recognized that different root traits play a significant role in withstanding stress conditions in many plant species, such as chickpea [14], common bean [15], soybean [16], rice [17], wheat [18], and maize [19]. For a new green revolution, the root morphological traits of crops have become a target for increasing crop yield and quality [20]. These studies suggest the importance of root morphological traits in crop growth and development.

In conventional breeding, crop improvement is achieved by phenotypic selection, whereby breeders monitor better progeny lines to achieve genetic improvement with target traits [21]. However, in modern times, interspecific crossing is used for genetic improvement, as desired traits are transferred from wild relatives to cultivated species [22]. Naturally, soybean varieties derived from *G. soja* are a vital wellspring of rich genetic assets, teeming with an abundance of unique alleles [23,24]. Therefore, next-generation sequencing of the wild soybean is crucial for genetic research and breeding [25,26]. In addition, primary features, such as overall root length, volume, and surface area, of wild soybean accessions are relatively smaller than those of commodity soybeans [27,28]. Therefore, it is essential to research root features and the interactions between roots and their environments [29]. In soybean breeding programs, phenotypic traits were taken using morphological features, such as visual root score assistance, for the easiest phenotypic evaluation of soybeans [30].

The regulation of root morphology is a multifaceted process influenced by a combination of genetic and environmental factors [31,32]. To date, several quantitative trait loci (QTL) have been identified and reported for the morphological traits of soybean roots [27,29,33,34,35]. Consequently, they are commonly employed in genetic research endeavors to enhance our understanding of the genetic constituents of the morphological traits of roots. Several QTLs with alleles from wild soybeans have been identified and reported to have various root morphological traits [27,28]. Similarly, a number of genes associated with root development, such as *Glyma.07g126400*, *Glyma.07g127300*, *Glyma.07g127100*, *Glyma.08g12320*, *Glyma.08g121770*, *Glyma.08g09550*, *Glyma.08g13900*, and *Glyma.16g141800*, have been found to enhance and control the root morphological traits in an interspecific mapping population of soybean [27,28,35]. Interspecific mapping root studies are relatively underrepresented; therefore, we focused on an interspecific soybean root study. Additionally, the measurement of root morphological traits was very difficult and laborious for many genotypes, and high phenotypic variations were observed in field conditions due to numerous factors, including soil density, distribution of nutrients, and water content [36,37]. Therefore, researchers have developed several software programs for root analyses, such as the SmartRoot (semi-automated image analysis, https://smartroot.github.io/SmartRoot-Installation/, accessed on 11 May 2023) software [38]. In our experiment, we used WinRHIZO Pro version 2019 computer-based image analysis software, which can easily measure root morphological traits, including root length, diameter, and volume.

In this study, we used an interspecific mapping population derived from a crossing between cultivated soybean ‘Williams 82’ and wild soybean accession ‘PI366121’ and used 3K single-nucleotide polymorphism (SNP) markers for linkage map construction. The main objectives of this study were to detect the major QTLs and determine the potential genes underlying the significant genomic regions of soybean seedling root morphological traits in an interspecific mapping population.

## 2. Results

### 2.1. Phenotypic Analysis of Root Traits

Table 1 shows the phenotypic measurements of root characteristics in parental genotypes and recombinant inbred lines (RILs). The descriptive phenotypic data are displayed in normal frequency distributions as average diameter (AD), root volume (RV), and link average diameter (LAD) (Figure 1(A1–A3,B1–B3,C1–C3)). In environment 1 planting, the average AD was evenly distributed across the population, with values of 0.33–0.50 mm and a mean value of 0.37 mm. For RV and LAD, we obtained values of 0.28–3.54 cm^3^ and 0.38–0.57 mm with corresponding mean values of 1.34 cm^3^ and 0.45 mm, respectively. For environment 2, the values of AD, RV, and LAD were uniformly distributed across the population, with values of 0.31–0.68 mm, 0.30–2.40 cm^3^, and 0.37–0.87 mm, respectively, and mean values of 0.36 mm, 1.24 cm^3^, and 0.46 mm, respectively. However, compared with the parents, transgressive segregation was found in AD, RV, and LAD root traits in both environments (Table 1). The coefficient variation (CV) of the root traits was comparable in different environments. Among these traits, CV was the largest for RV (31.10%), followed by AD (13.57%), and LAD (13.78%) in the first environment. A similar CV trend was found in environment 2. Most skewness values ranged from 0.52 to 1.08 in all root traits in environment 1; however, in environment 2, skewness values were 4.64, 0.18, and 4.11 for AD, RV, and LAD, respectively. A negative kurtosis value was observed for the RV (−0.22) root trait in environment 2. Therefore, based on the skewness and kurtosis values, all three root traits were normally distributed (Figure 1). 

An analysis of variance (ANOVA) for the three root traits was performed to understand the genotype, environment, and their interaction for the traits. Significant variations were observed for genotypes and the interaction between genotype and environment (*p* < 0.0001) in all root traits (Table 2). Pearson’s correlation coefficient test showed that there was a significant positive correlation between any two root traits in both environments (Table 3). A significant positive correlation was observed between AD and LAD (*r* = 0.97, *p* < 0.0001), followed by RV and LAD (*r* = 0.56, *p* < 0.0001) in environment 1. Similar results were observed for environment 2. 

### 2.2. QTL Detection of Root Traits

A total of 42 significant QTLs were detected on 12 different chromosomes (chr.), including 17 for AD, 9 for RV, and 16 for LAD; some were co-located QTLs in this mapping population (Table 4 and Figure 2). Among these QTLs, 12 were considered major QTLs, as their phenotypic variation was >30% [39]. 

For AD, 17 QTLs were identified on chr. 2, 3, 4, 8, 12, 13, 17, 18, 19, and 20. Each of these 17 QTLs contributed to phenotypic variance (*R*^2^) ranging from 7.8% to 39.1% and a logarithm of odds (LOD) value of 2.7–18.7. One of the 17 QTLs, *qAD_D2-19-1*, had an LOD value of 18.7 and a phenotypic variation of 39.0%. The wild soybean parent ‘PI366121’ contributed a positive allele on chr. 2, 4, 8, 12, 13, 19, and 20 for AD; the cultivated soybean, Williams 82, contributed a positive allele on chr. 3, 18, 19, and 20 for the trait (Table 4 and Figure 2).

Nine QTLs for RV were found on chr. 2, 3, 4, 10, 13, and 17, with LOD scores of 2.6–4.4. They had a phenotypic variation of 5.5–16.5%. The *qRV_D1-3-1* QTL was on chr. 3, accounting for a phenotypic variation of 16.5% with an LOD value of 4.4 and a positive allele from cultivated soybean ‘William 82’ (Table 4 and Figure 2).

For LAD, 16 QTLs were detected on chr. 3, 4, 8, 13, 15, 17, 18, 19, and 20. The phenotypic variation and LOD score ranged from 6.3% to 36.2% and 2.7 to 16.4, respectively. High LOD values were found for the QTLs *qLAD_D2-19-1* (16.4), *qLAD_D2-8-1* (15.0), *qLAD_D2-3-1* (14.9), *qLAD_D2-13-1* (13.8), and *qLAD_D2-20-1* (12.0). *qLAD_D2-19-1,* flanked by Gm19_42673649_A_C and Gm19_50184509_A_G markers, accounted for 36.1% of the phenotypic variation. Favorable alleles of this trait came from both cultivated and wild parents (Table 4 and Figure 2).

SNP markers at 244.8 cM in the interval of Gm3_829023_G_T-~3365988_T_G and SNP markers at 24.7 cM in the interval of Gm13_27527083_G_T~43496306_A_G were associated with AD, RV, and LAD (Figure 2). Moreover, QTL regions for AD and LAD were detected on chr. 8, 18, 19, and 20 at positions 72.0, 64.2, 85.7, and 4.0 cM, respectively (Table 4). 

### 2.3. Putative Candidate Genes and Gene Expression in QTL Regions

Putative candidate genes were identified within the two most hotspot QTL regions on chr. 3 and 13 (Appendix A). A total of 198 putative candidate genes detected for the most effective SNPs associated with root traits and genes are presented in Appendix A. Among the 198 genes, 89 genes had allele variations between the resequencing data of ‘William 82’ and ‘PI366121’ (Appendix A). Analysis of the sequence of the parental line ‘PI366121’ [40] within the QTL regions on chr. 3 and 13, 32 genes revealed missense and splice variants that cause amino acid changes (Table 5 and Appendix A). Moreover, tissue-specific transcriptome data of roots, root stripped, root tips, root hairs, leaves, meristem, green pod, and root nodules were downloaded from the ePlant soybean database (https://bar.-utoronto.ca/eplant soybean/, accessed on 14 June 2023), and used to assess the expression of the candidate genes (Appendix A). Based on transcriptome data, six genes having high expression in root tissues: *Glyma.03G027500* (transketolase/glycoaldehyde transferase), *Glyma.03G014500* (dehydrogenases), *Glyma.13G341500* (leucine-rich repeat receptor-like protein kinase), *Glyma.13G341400* (AGC kinase family protein), *Glyma.13G331900* (60S ribosomal protein), and *Glyma.13G333100* (aquaporin transporter) (Table 6, Figure 3). Among the six candidate genes, three genes (*Glyma.03G014500*, *Glyma.13G341400*, and *Glyma.13G331900*) exhibited the highest expression in root tissue compared with other tissues (leaf, seedling, shoot, stem, meristem, flower, pod, nodule, seed, embryo, and endosperm) using RNA-Seq soybean libraries (4085) (http://ipf.sustech.edu.cn/pub/soybean/, accessed on 22 June 2023). Appendix A shows the differential expression levels of the three candidate genes in the other tissues. 

## 3. Discussion

The interspecific mapping population used in this study was first genotyped using the GoldenGate^®^ assay, which contained 1536 SNP loci with 169 F_4:5_ RILs derived from ‘Williams 82’ × ‘PI366121’ [41]. The genetic map was constructed with 414 polymorphic and filtered SNPs to detect the QTL regions for foxglove aphid resistance [42,43], 100-seed weight [44], and seed starch content [40]. In this study, a single plant was randomly selected from 157 F_10_ RILs to increase homozygosity and genotyped using the BARCSoySNP3K SNP array, which contained a subset of 2830 SNP loci distributed across 20 soybean chromosomes from BARCSoySNP6K [45]. In this study, >1400 SNPs were used to construct the improved genetic map, resulting in a better estimation of QTL positions for the linkage mapping study with this interspecific population. 

The root is the hidden part of a plant that plays a crucial role in plant development by serving as a stabilizer, absorbing water and nutrients, and interacting with the soils microorganisms. Our study employed an interspecific crossing between ‘Williams 82’ and ‘PI366121’ and produced an RIL population to map QTLs for root traits. Parents and the RILs showed wide phenotypic variation for root attributes. In contrast to both parents, the cultivated soybean ‘Williams 82’ has a well-developed root system compared with the wild soybean ‘PI366121’, and transgressive segregation was observed in this study (Table 1 and Figure 4A–E). A rice QTL, *DEEPER ROOTING1* (*DRO1)*, regulates the root system architecture, including root angle and root tips, and increases rice yield [46]. Several studies have reported that root traits, including root length, surface area, and volume, have been found on different chromosomes, including chr. 3, 7, 8, and 20, which improve seedling growth in soybean [27,28,29]. Fibrous root-related QTLs have been observed on chr. 3, 4, 8, and 20 [34]. Furthermore, average root diameter, lateral root number, and RV-related QTLs were found on chr. 7, 17, and 20, which enhanced soybean root growth and development [47,48]. 

In this study, we found nine important genomic regions on 12 chromosomes that were strongly linked with root morphological traits (Table 4). In more than 60% of the total root trait QTLs, a positive allele was contributed by the wild soybean accession ‘PI 366121’, with phenotypic variations of 6.9–39.0%, which significantly affected root attributes. Remarkably, AD QTL was mapped in this population on chr. 3 with positive alleles contributed by the cultivated soybean ‘William 82’ and co-located with LAD and RV QTLs, accounting for phenotypic variations of 16.2–39.1%. A positive allele contributed by inferior parental lines of both wild and cultivated soybeans has been reported in some root trait-related studies [28,49]. In our genomic regions, 12 QTLs had phenotypic variations of >30% and were considered major QTLs [39]. QTL *qAD_19-1* was found on chr. 19 at position 85.6 cM, accounting for 39.0% phenotypic variation. Similar root traits of *Qrd 14-1* and *Qrd 12-1* were reported with 15% and 22% phenotypic variation and 3.36 and 6.15 LOD scores, respectively [50], and a single AD QTL was found with 7% phenotypic variation [49].

Based on our results, two QTL groups were mapped on chr. 3 and 13, along with co-located QTLs for AD, RV, and LAD (Appendix A). *qAD-3*, *qRV-3*, *qLAD-3*, *qAD-13*, *qRV-13*, and *qLAD-13* were located in the same intervals on chr. 3 and 13, with a wide phenotypic variation (Table 4). The six QTLs may regulate soybean seedling root morphology, such as AD, RV, and LAD, as well as root development. Surprisingly, AD, RV, and LAD-related QTLs on chr. 3 have not been found in any other root studies. However, other root traits such as root length, distribution, and score-related QTLs were found in chr. 3 [28,29,34]. In chr. 13, only root weight QTL was found [29]; however, AD, RV, and LAD-related QTLs have not been reported yet. Additionally, some co-located QTLs were observed for soybean mapping populations, such as seedling root [27], plant height [51,52], and leaf morphology [53]. A significantly positive correlation was detected between AD and LAD (*r* = 0.97), followed by RV and LAD (*r* = 0.56) and AD and RV (*r* = 0.55) in our study, influencing root growth, development, and phenotypic variation of soybean seedlings. In the early growth stage of soybean, the total root length and photosynthetic efficiency correlated positively, influencing the lateral and fibrous rooting capacity under limited water conditions [54]. A weak positive correlation was observed between AD and RV (*r* = 0.32) and LAD and RV (*r* = 0.34) in soybean [49].

The SNP markers associated with root features, including AD, RV, and LAD, on chr. 3 and 13, were within the same marker intervals. This genomic region may harbor some root-functioning genes, thereby prompting the root growth and development of soybean seedlings. Soybean root length in chr. 3 with marker interval NCSB_000550–SNP5617_Magellan was mentioned as a potential candidate gene [28]. Several researchers have reported that the maker regions improve soybean root development and stress resistance [55,56,57]. Furthermore, root architectural traits minimize the drought tolerance of rice [17,57], and marker-assisted breeding improves the root system architecture of maize [58].

In this study, we performed analyses on an interspecific mapping population along with the entire genome transcriptome data of soybeans to identify genes associated with potential root traits and examine their differences in tissue/organ-specific expression. Based on our results, 32 genes for SNP variations were identified, including missense and splice variants that altered amino acids (Table 5). A similar method was reported, and variants were observed in soybean root traits [16,27], meristems in *Arabidopsis* [59], and lateral roots in rice [60]. In our study, the transcriptome data were used for gene expression with different tissue-specific expressions (Figure 3). A similar tactic was reported, and candidate genes for soybean root traits were identified [48]. We identified five candidate genes, including *Glyma.03G027500* (transketolase/glycoaldehyde transferase), *Glyma.03G014500* (dehydrogenases), *Glyma.13G341500* (leucine-rich repeat receptor-like protein kinase), *Glyma.13G341400* (AGC kinase family protein), and *Glyma.13G331900* (60S ribosomal protein), which showed greater expression in the root. Compared with other plant tissues, these genes may enhance soybean seedling root morphological traits as well as AD, RV, and LAD. The leucine-rich repeat protein (LRP) class was observed within the root QTL of soybean, significantly influencing other root properties, including the number of lateral roots, root diameter, and RV [28]. Furthermore, LRP regulates *Medicago truncatula* roots [61]. The AGC protein kinase regulates auxin transport polarity [62] and organ (root) development in *Arabidopsis* [63]. The 60S ribosomal protein L14-2 reportedly enhances the drought and salt tolerance of cotton [64]. Additionally, we found one aquaporin functioning gene, *Glyma.13G333100* (Aquaporin Transporter), which probably induces root development and enhances AD and RV. The same gene was reported in soybeans, which regulated seedling root growth under heat stress [65], and the same gene ID *AT4G01740* (*TIP1;3*) controlled drought stress in *Arabidopsis* [66]. Six potential genes were recently identified in seven root traits, including AD, LAD, and the number of tips in soybean landraces [67]. Therefore, we conclude that the common QTL regions will benefit breeders in improving soybean root morphological traits as well as AD, RV, and LAD.

## 4. Materials and Methods

### 4.1. Plant Materials and Growth Conditions

In this study, an interspecific mapping population was used. A population of 157 F_10:11_ RILs was developed from a crossing between cultivated soybean ‘Williams 82’ and wild soybean ‘PI366121’ [42]. These RILs and parental lines were grown in polyvinyl chloride (PVC) pipes (6 cm (diameter) × 40 cm (height)). Furthermore, all (636) PVC pipes were placed on 16 trays (35 cm (wide) × 65 cm (long)) in a greenhouse at the research center of the Kyungpook National University, Daegu, South Korea. Each tray contained 40 PVC pipes; the trays, including the parents ‘Williams 82’ and ‘PI366121’, served as a control. Sandy soil was used in our experiment. Two seeds of each parent and RIL population were planted in the PVC pipes on 25 March 2022 (environment 1) and 27 April 2022 (environment 2), respectively. The study was conducted in a completely randomized block design (RCBD) with three replications. Greenhouse photoperiod (14 h daytime) and temperature (28 °C ± 2 °C) were maintained for seedling growth. After the germination of seeds, they were thinned, and a single seedling was allowed to grow for root analysis. At 24 days after the third trifoliate leaf developed, i.e., at the V3 stage, the seedlings were harvested for root samples in both environments. 

### 4.2. Root Phenotypic Evaluation

Root phenotypic parameters were evaluated at the V3 stage of soybean seedlings in different environments (Figure 5A). During harvesting, all sandy soil was removed from the PVC pipes very carefully, and root samples were separated from the soil. The root samples were then softly washed with clean tap water and kept in medium (20 cm long × 15 cm wide) plastic bags containing a small amount (15–20 mL) of water to maintain moisture in the samples (Figure 5D). Next, a scanner was used to capture clear 2D root images (Epson, Expression 12,000XL, Nagano, Japan). A transparent plastic tray (30 cm long × 20 cm wide), which contained normal clean water, was used for root sample scanning (Figure 5E). The scanned root images were analyzed using WinRHIZO Pro software version 2019 (Regent Instruments Inc., Québec City, QC, Canada) (Figure 5F). In this study, three root trait parameters, AD, RV, and LAD, were measured (Table 7). LAD was considered a minor root trait. According to the WinRHIZO description (https://regent.qc.ca/assets/winrhizo_software.html accessed on 4 June 2023), LAD is a link of the root part between two forks or a fork and a tip. It is a study of the morphology and basic interaction of root segments measured by the AD of links that belong to an order. Therefore, we hypothesized that LAD affected AD and helped in root development by uptaking water and nutrients [10].

**Table 7 ijms-25-04687-t007:** List of soybean root morphological traits evaluated in the soybean mapping population.

**Trait Abbreviation**	**Description (Units)**
AD	Average diameter (mm)
RV	Root volume (cm^3^)
LAD	Link average diameter (mm)

### 4.3. Genotyping and Linkage Map Construction

The single plant of 157 F_10_ RILs and their parental lines (‘Williams 82’ and ‘PI366121’) were randomly selected, and genomic DNA from leaves were genotyped using the BARCSoySNP3K SNP array, which contained a subset of 2830 SNP loci distributed across 20 soybean chromosomes from BARCSoySNP6K [45]. This SNP genotype was developed at the soybean genomics and improvement laboratory at the USDA. After filtering, 1408 of the 2830 SNP markers were polymorphic between the parents and were used to construct a genetic linkage map for the 20 linkage groups. The SNP markers were binned based on their segregation patterns among the RIL population, employing the bin function in IciMapping 4.2 (http://www.isbreeding.net, accessed on 4 June 2023) [68]. Using the Kosambi mapping function, bin markers were systematically grouped and categorized using IciMapping 4.2 [69]. The total length of the soybean mapping population map was 4426.7 cM, with an average distance between the adjacent markers of 3.14 cM (Appendix A). The average length of individual chromosomes or linkage groups was 221.33 cM, with an average number of markers per linkage group of 70.4 (Appendix A). MapChart 2.2 (http://www.biometris.nl/uk/Software/MapChart/, accessed on 10 June 2023) software was used to draw the genetic linkage map [70]. Genomic DNA was isolated from the fresh soybean young leaves using a modified cetyl trimethylammonium bromide method [71].

### 4.4. QTL Mapping Analysis

The QTLs for soybean root traits were identified using WinQTLCart 2.5 based on the composite interval mapping procedure [72]. The software program uses an enhanced algorithm for composite interval mapping, which has a heightened ability to identify QTLs, lower false detection rates, and provide less skewed QTL effect estimates. The model 6 was used, and the window size was set at 10 cM with background cofactors. To declare significant QTLs with a more rigorous LOD threshold, a permutation (*p* = 0.05) test was performed with 1000 runs for the mapping software packages for all traits. The forward regression approach was employed to determine the walking speed, which was set at 2 cM. The QTL analysis incorporated the mean values of the three distinct soybean root morphological traits. The QTL map positions on the linkage maps were depicted using the Mapchart program. 

### 4.5. Candidate Gene Identification and Expression Analysis

We used the two most significant QTL regions (Table 4 and Appendix A) on chr. 3 and 13 to identify potential candidate genes based on annotation using Soybase (https://soybase.org/SequenceIntro.php, accessed on 5 July 2023) according to the ‘Wm82.a2.v1’ soybean reference genome. The genome browser was used to identify potential genes within each significant SNP flanking region, and the Phytozome database [73] was used to perform functional annotation of the genes (Appendix A). Furthermore, we identified the SNP and INDEL variations with genomes ‘Williams 82’ and ‘PI366121’, using the Soykb database (https://soykb.org/SNPViz2/, accessed on 18 July 2023) [74] (Appendix A). We used the ePlant soybean database (https://bar.utoronto.ca/eplant-soybean/, accessed on 18 July 2023) to identify the root trait-related potential gene expression for putative candidate genes, and a heatmap was constructed with root tissue data by the Tbtools software (https://github.com/CJ-Chen/Tbtools, accessed on 29 July 2023) (Appendix A). The selected genes were used for distinction expression analysis in diverse tissues using web-based, publicly available RNA-Seq soybean library data (4085) with default settings (http://ipf.sustech.edu.cn/pub/soybean/, accessed on 21 July 2023) [75]. 

### 4.6. Statistical Analysis

The experiment used a randomized complete block design (RCBD) with three replications. The root phenotypic traits of the parental lines and RIL population were tested for descriptive statistics with a normal frequency distribution using IBM SPSS statistics 25. To assess statistical significance, we performed an ANOVA with SAS (SAS release 9.4; SAS, Gary, NC, USA). Correlation (Pearson correlation coefficients) analysis was performed using SAS PROC CORR to determine the relationship between root traits. In Soybase, the chr. numbers corresponded to the soybean genetic linkage group (http://www.soybase.org, accessed on 20 April 2023).

## 5. Conclusions

In this study, we performed QTL mapping for soybean root traits in an interspecific population derived from the ‘Williams 82’ and ‘PI366121’ crosses. We identified 42 QTLs, including 12 major QTLs, distributed across 12 chromosomes. The QTL regions on chr. 3 and 13 were found to control AD, RV, and LAD, with positive alleles derived from both ‘PI366121’ and ‘Williams 82’. These QTL regions have the potential to enhance root development. Additionally, we detected six candidate genes within the most significant QTL regions that influence soybean root morphological traits. In conclusion, these significant regions are valuable for breeders seeking to improve soybean root morphological traits, leveraging alleles inherited from wild soybean accessions.

## Figures and Tables

**Figure 1 ijms-25-04687-f001:**
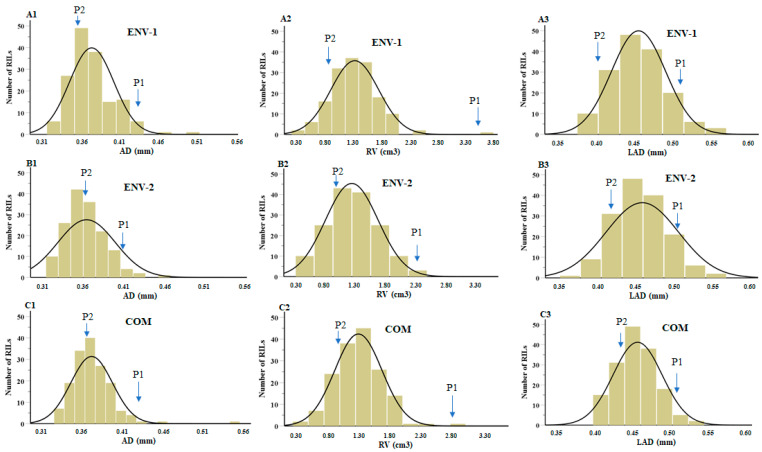
Frequency distribution with normal curve of soybean root traits in different environments (ENV-1, ENV-2, and COM) of mapping populations. (**A1**–**A3**) AD, RV, and LAD for ENV-1. (**B1**–**B3**) AD, RV, and LAD for ENV-2. (**C1**–**C3**) AD, RV, and LAD for COM. The traits are described in Table 7. Arrows represent the mean values of P1 (‘William 82’) and P2 (‘PI366121’). ENV: environment; COM: combined.

**Figure 2 ijms-25-04687-f002:**
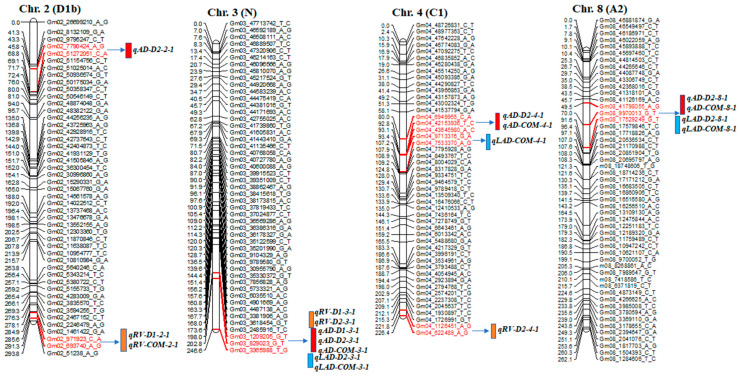
Diagram showing the location of root QTLs on 12 different chromosomes: Chr. 2, 3, 4, 8, 10, 12, 13, 15, 17, 18, 19, and 20 detected in an interspecific mapping population of soybean. Genetic distance and markers are on the left side of the linkage groups, and marker names are shown on the right side of the linkage groups. Colored bars indicate QTL regions.

**Figure 3 ijms-25-04687-f003:**
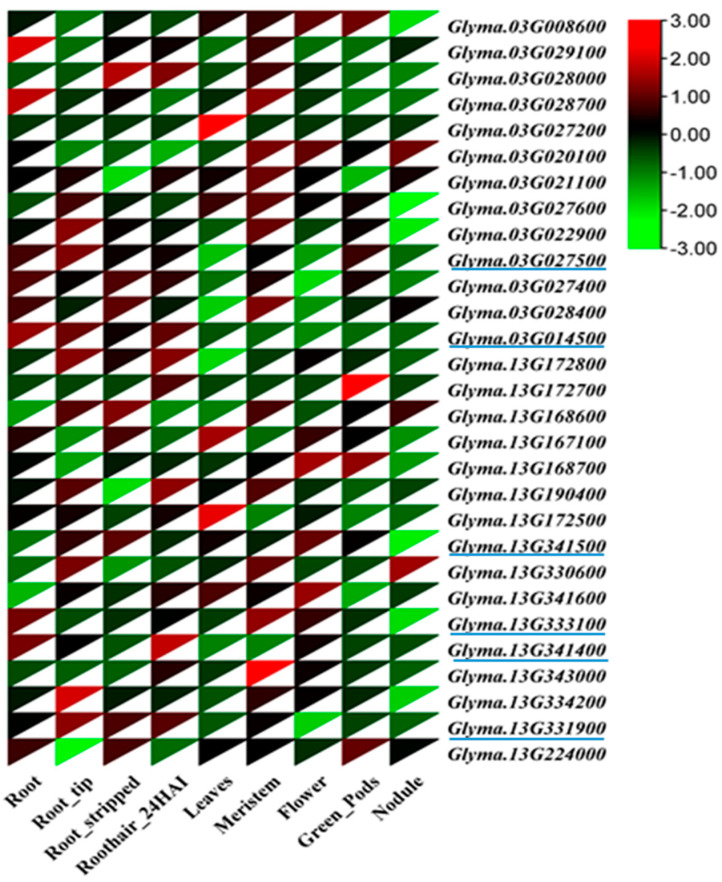
Heatmap showing the expression level of candidate genes in different plant tissues, including soybean roots. The red color indicates a high expression greater than 0, and the green color indicates a low expression less than 0 among the parameters Blue color underline indicates the final candidate genes.

**Figure 4 ijms-25-04687-f004:**
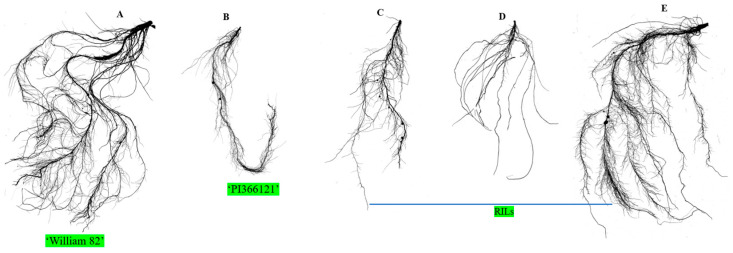
Variation in 2D root image of the morphology of soybean seedlings; five randomly selected root samples, including parents (**A**) ‘William 82’, (**B**) ‘PI366121’, and (**C**–**E**) RILs.

**Figure 5 ijms-25-04687-f005:**
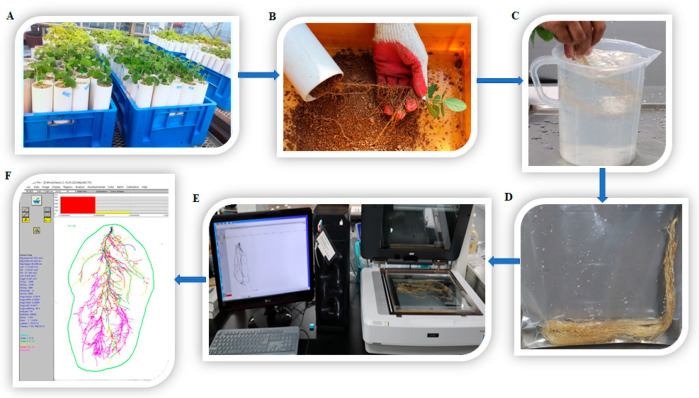
Procedure for soybean seedling root analysis using WinRHIZO software. (**A**) Harvesting at the V3 stage of soybean seedlings. (**B**) Removal of soil and separation of roots from the pipe. (**C**) Washing of the root sample. (**D**) Clean roots are kept in plastic bags with a small amount of water. (**E**) Scanning of the clean root sample. (**F**) Analysis of the root sample.

**Table 1 ijms-25-04687-t001:** Descriptive statistics for the three root traits of the soybean interspecific mapping population.

Traits	Parents		RIL Population			Range	CV	SD	Skewness	Kurtosis
	William 82	PI 366121	Mean	Min	Max					
ENV-1										
AD	0.43	0.36	0.37	0.33	0.50	0.18	13.57	0.03	1.08	2.30
RV	3.54	1.10	1.34	0.28	3.54	3.27	31.10	0.42	0.88	4.32
LAD	0.53	0.44	0.45	0.38	0.57	0.19	13.78	0.03	0.52	0.20
ENV-2										
AD	0.41	0.37	0.36	0.31	0.68	0.37	11.56	0.03	4.64	3.18
RV	2.31	1.13	1.24	0.30	2.40	2.11	34.99	0.43	0.18	−0.22
LAD	0.53	0.48	0.46	0.37	0.87	0.49	11.76	0.04	4.11	8.02
COM										
AD	0.42	0.37	0.37	0.32	0.54	0.22	9.36	0.02	2.48	4.38
RV	2.93	1.11	1.30	0.29	2.93	2.63	28.89	0.37	0.53	1.81
LAD	0.53	0.46	0.46	0.40	0.69	0.34	12.28	0.03	2.36	4.40

CV: coefficient of variation; ENV: environment; COM: combined; Min: minimum; Max: maximum; SD.: standard deviation.

**Table 2 ijms-25-04687-t002:** F-value from analysis of variance for the three root traits of the interspecific mapping population.

Source	AD	LAD	RV
GEN	4.77	3.71	16.04
	*p* < 0.0001	*p* < 0.0001	*p* < 0.0001
ENV	27.16	3.43	39.42
	<0.0001	0.065	*p* < 0.0001
REP	2.51	3.79	0.46
	0.114	0.052	0.499
GEN*ENV	2.97	2.26	4.84
	*p* < 0.0001	*p* < 0.0001	*p* < 0.0001

The traits are described in Table 7. The bottom value in each cell indicates the significance level. GEN: genotype; ENV: environment; REP: replication; *: interaction.

**Table 3 ijms-25-04687-t003:** Phenotypic Pearson correlation coefficient among the three soybean root traits.

Traits	AD	LAD	RV
AD		0.97	0.55
		*p* < 0.0001	*p* < 0.0001
LAD	0.98		0.56
	*p* < 0.0001		*p* < 0.0001
RV	0.39	0.43	
	*p* < 0.0001	*p* < 0.0001	

The environment 1 data are presented in the upper right triangular matrix of Table 3, whereas the environment 2 data are presented in the lower left triangular matrix of Table 3. The traits are described in Table 7, where *p* is the significant level.

**Table 4 ijms-25-04687-t004:** QTLs for the soybean root traits identified using composite interval mapping in an interspecific mapping population.

Trait	QTL Name	Chr.	Left and Right Markers	Position	LOD	*R*^2^ (%)	Add
AD_D2	*qAD-D2-2-1*	2	02_7790424_A_G~51272051_C_A	47.7	13.8	39.0	−0.15
RV_COM	*qRV-COM-2-1*	2	02_693740_A_G~971923_C_A	289.6	2.9	6.9	0.10
RV_D1	*qRV-D1-2-1*	2	02_693740_A_G~971923_C_A	289.6	2.8	6.5	0.11
AD_D2	*qAD-D2-3-1*	3	03_829023_G_T~3365988_T_G	244.8	17.2	39.1	0.15
LAD_D2	*qLAD-D2-3-1*	3	03_829023_G_T~3365988_T_G	244.8	14.9	36.2	0.19
LAD_COM	*qLAD-COM-3-1*	3	03_829023_G_T~3365988_T_G	244.8	6.4	20.9	0.07
AD_COM	*qAD-COM-3-1*	3	03_829023_G_T~3365988_T_G	244.8	5.9	21.2	0.04
RV_D1	*qRV-D1-3-1*	3	03_829023_G_T~3365988_T_G	244.8	4.4	16.5	0.78
AD_D1	*qAD-D1-3-1*	3	03_829023_G_T~3365988_T_G	240.8	2.7	27.2	0.02
RV_D2	*qRV-D2-3-1*	3	03_829023_G_T~1209205_G_T	200.0	2.6	6.9	−0.12
AD_D2	*qAD-D2-4-1*	4	04_6949955_C_A~42153936_T_C	92.0	12.4	39.0	−0.15
LAD_COM	*qLAD-COM-4-1*	4	04_7533370_A_G~9713316_G_A	95.4	5.9	24.4	−0.10
AD_COM	*qAD-COM-4-1*	4	04_43645980_A_C~9713316_G_A	93.4	3.1	14.5	−0.04
RV_D2	*qRV-D2-4-1*	4	04_622489_A_G~1126451_A_G	223.8	2.8	7.3	0.12
AD_D2	*qAD-D2-8-1*	8	08_9970013_G_T~17529249_G_T	72.0	17.4	39.0	−0.15
LAD_D2	*qLAD-D2-8-1*	8	08_9970013_G_T~17529249_G_T	72.0	15.0	36.1	−0.19
LAD_COM	*qLAD-COM-8-1*	8	08_9970013_G_T~17529249_G_T	72.0	7.0	24.4	−0.10
AD_COM	*qAD-COM-8-1*	8	08_41798055_A_G~10330658_A_C	69.5	6.4	22.6	−0.08
RV_D2	*qRV-D2-10-1*	10	10_43840376_T_C~44574663_C_T	95.1	2.7	6.3	0.11
AD_D2	*qAD-D2-12-1*	12	12_6195964_C_T~688182_T_G	264.7	3.6	19.6	−0.04
AD_D2	*qAD-D2-13-1*	13	13_27527083_G_T~42337548_A_C	24.7	16.2	39.0	−0.15
LAD_D2	*qLAD-D2-13-1*	13	13_27527083_G_T~42337548_A_C	24.7	13.9	36.1	−0.19
LAD_COM	*qLAD-COM-13-1*	13	13_27527083_G_T~42337548_A_C	22.7	4.5	17.5	−0.06
AD_COM	*qAD-COM-13-1*	13	13_27527083_G_T~42337548_A_C	22.7	3.0	14.5	−0.04
RV_D2	*qRV-D2-13-1*	13	13_42337548_A_C~43496306_A_G	40.0	2.9	8.3	0.15
LAD_COM	*qLAD-COM-15-1*	15	15_50864537_A_G~14712034_G_T	179.6	6.4	24.4	−0.10
AD_D2	*qAD-D2-17-1*	17	17_910402_T_G~7756014_C_T	3.3	13.2	39.0	−0.15
LAD_COM	*qLAD-COM-17-1*	17	17_8449684_G_A~7794179_A_G	0.3	3.3	17.5	−0.06
RV_COM	*qRV-COM-17-1*	17	17_4967175_G_A~8109237_A_C	130.3	2.6	5.7	0.09
RV_D1	*qRV-D1-17-1*	17	17_4967175_G_A~8109237_A_C	130.3	2.6	5.5	0.11
LAD_D1	*qLAD-D1-18-1*	18	18_61963157_A_G~62036271_A_G	0.0	4.6	11.0	−0.02
LAD_D1	*qLAD-D1-18-2*	18	18_52751146_G_A~53762458_A_G	64.2	4.5	10.8	0.01
AD_D1	*qAD-D1-18-1*	18	18_52751146_G_A~53762458_A_G	64.2	3.7	9.1	0.01
LAD_COM	*qLAD-COM-18-1*	18	18_52751146_G_A~53762458_A_G	64.2	3.4	8.4	0.01
AD_COM	*qAD-COM-18-1*	18	18_52751146_G_A~53762458_A_G	64.2	3.3	7.8	0.01
AD_D2	*qAD-D2-19-1*	19	19_42673649_A_C~50184509_A_G	85.7	18.7	39.0	−0.15
LAD_D2	*qLAD-D2-19-1*	19	19_42673649_A_C~50184509_A_G	85.7	16.4	36.1	−0.19
LAD_COM	*qLAD-COM-19-1*	19	19_42673649_A_C~50184509_A_G	85.7	8.4	24.4	−0.10
AD_COM	*qAD-COM-19-1*	19	19_42673649_A_C~50184509_A_G	85.7	7.6	22.6	−0.08
LAD_D2	*qLAD-D2-20-1*	20	20_26500747_G_A~43146832_A_G	7.2	12.0	36.1	−0.19
LAD_COM	*qLAD-COM-20-1*	20	20_32603292_A_G~38578470_T_C	4.0	5.1	17.5	−0.06
AD_COM	*qAD-COM-20-1*	20	20_32603292_A_G~38578470_T_C	4.0	3.7	14.6	−0.04

The traits are described in Table 7. Chr.: chromosome; D1: environment 1; D2: environment 2; COM: combined; LOD: logarithm of the odds; R^2^: phenotypic variation explained; Add: additive effects. The positive sign (+) indicates alleles contributed by ‘William 82’ and the negative sign (−) by ‘PI366121’.

**Table 5 ijms-25-04687-t005:** SNPs of annotated genes on chromosomes 3 and 13 based on genomic information of ‘William 82’ and ‘PI366121’.

SNP Position	Gene Name	Ref.	Alt.	Mutation	Start	End	Strand
Chr03:832559	*Glyma.03G008600*	G	A	Splice region variant	830,451	838,408	−
Chr03:1393538	*Glyma.03G013700*	G	T	Missense variant	1,393,484	1,393,734	+
Chr03:1452530	*Glyma.03G014500*	C	T	Missense variant	1,452,354	1,452,677	−
Chr03:2061229	*Glyma.03G020100*	A	C	Missense variant	2,057,797	2,063,029	−
Chr03:2164039	*Glyma.03G021100*	C	T	Splice region variant	2,162,406	2,167,712	+
Chr03:2413616	*Glyma.03G022900*	C	T	Missense variant	2,413,172	2,413,671	−
Chr03:2998840	*Glyma.03G027200*	A	G	Missense variant	2,997,332	2,999,026	+
Chr03:3010747	*Glyma.03G027400*	C	T	Missense variant	3,010,213	3,018,498	−
Chr03:3024508	*Glyma.03G027500*	G	A	Missense variant	3,024,204	3,024,656	−
Chr03:3029835	*Glyma.03G027600*	A	G	Splice region variant	3,029,154	3,033,239	−
Chr03:3038816	*Glyma.03G027800*	T	C	Missense variant	3,038,542	3,039,018	−
Chr03:3072041	*Glyma.03G028000*	G	C	Missense variant	3,071,857	3,072,066	−
Chr03:3109886	*Glyma.03G028400*	T	C	Missense variant	3,107,378	3,110,786	+
Chr03:3128983	*Glyma.03G028700*	A	G	Missense variant	3,128,776	3,130,648	+
Chr03:3178665	*Glyma.03G029100*	G	C	Missense variant	3,178,307	3,182,074	+
Chr13:28182349	*Glyma.13G167100*	T	A	Missense variant	28,182,121	28,182,399	+
Chr13:28303515	*Glyma.13G168600*	G	A	Missense variant	28,298,272	28,304,199	−
Chr13:28305241	*Glyma.13G168700*	C	T	Splice donor variant	28,305,098	28,310,222	+
Chr13:28645128	*Glyma.13G172500*	C	T	Missense variant	28,644,819	28,649,188	−
Chr13:28672639	*Glyma.13G172700*	T	A	Missense variant	28,671,831	28,675,231	+
Chr13:28686299	*Glyma.13G172800*	G	C	Missense variant	28,683,493	28,686,774	+
Chr13:30403159	*Glyma.13G190400*	G	T	Missense variant	30,402,029	30,409,606	+
Chr13:33685460	*Glyma.13G224000*	T	A	Missense variant	33,684,833	33,686,902	+
Chr13:42523190	*Glyma.13G330600*	A	T	Missense variant	42,523,075	42,523,280	−
Chr13:42650050	*Glyma.13G331900*	A	G	Splice region variant	42,648,871	42,650,712	−
Chr13:42711332	*Glyma.13G333100*	G	A	Missense variant	42,711,023	42,711,397	+
Chr13:42797618	*Glyma.13G334200*	T	C	Missense variant	42,794,878	42,801,645	+
Chr13:43246090	*Glyma.13G340400*	G	A	Splice region variant	43,244,791	43,246,867	−
Chr13:43311794	*Glyma.13G341400*	C	A	Missense variant	43,311,148	43,312,308	+
Chr13:43326463	*Glyma.13G341500*	C	A	Missense variant	43,325,404	43,326,510	−
Chr13:43333456	*Glyma.13G341600*	T	C	Missense variant	43,331,375	43,335,735	−
Chr13:43447450	*Glyma.13G343000*	G	C	Missense variant	43,447,390	43,451,125	−

Ref: reference allele from ‘Williams 82’; Alt: alternative allele from ‘PI 366121’. (+) for sense of DNA and (−) for antisense of DNA.

**Table 6 ijms-25-04687-t006:** The six candidate genes have IDs, function annotations, and physical position.

Final Candidate Gene	Gene Description	Start Position (Wm82.a2.v1)	End Position (Wm82.a2.v1)
*Glyma.03G027500*	TRANSKETOLASE/GLYCO-ALDEHYDE TRANSFERASE	3,024,204	3,024,656
*Glyma.03G014500*	DEHYDROGENASES WITH DIFFERENT SPECIFICATIONS	1,452,354	1,452,677
*Glyma.13G331900*	60S RIBOSOMAL PROTEIN L35	42,648,871	42,650,712
*Glyma.13G341400*	AGC KINASE FAMILY PROTEIN	43,311,148	43,312,308
*Glyma.13G341500*	LEUCINE-RICH REPEAT RECEPTOR-LIKE PROTEIN KINASE	43,325,404	43,326,510
*Glyma.13G333100*	AQUAPORIN TRANSPORTER	42,711,023	42,711,397

## Data Availability

The datasets generated in this study are available from the corresponding author upon reasonable request.

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
