# Peer review of "Identification of Quantitative Trait Loci Controlling Root Morphological Traits in an Interspecific Soybean Population Using 2D Imagery Data"

_ijms, 2024, doi:10.3390/ijms25094687_

Round 1
Reviewer 1 Report
Comments and Suggestions for Authors
This paper used two-dimensional image data to identify quantitative trait loci (QTL) controlling root traits in an interspecific mapping population derived from a cross between wild soybean ‘PI366121’ and cultivar ‘Williams 82’. Forty-two QTLs were identified on twelve chromosomes, twelve of which were identified as major QTLs. Two significant QTL regions for the average diameter, root volume, and link average diameter root traits were detected on chromosomes 3 and 13, and both wild and cultivated soybeans contributed positive alleles. Six candidate genes Glyma.03G027500 (transketolase/glycoaldehyde transferase), Glyma.03G014500 (dehydrogenases), Glyma.13G341500 (leucine-rich repeat receptor-like protein kinase), Glyma.13G341400 (AGC kinase family protein), Glyma.13G331900 (60S ribosomal protein), and Glyma.13G333100 (aquaporin transporter) showed higher expression in root tissues based on publicly available transcriptome data. These results will help breeders improve soybean genetic components and enhance soybean root morphological traits using desirable alleles from wild soybean. Overall, this work is sound, interesting. I have several recommendations:
For the introduction, I suggest the author introduce several key genes that have been reported involved in controlling soybean root traits.
Figure 2 is hard to read.
In Figure 3, how to understand that the expression value of gene is less than zero?
Author Response
Dear Reviewer 1, we thank you for your valuable comments and suggestions for improving our manuscript. According to your suggestions we have made the necessary corrections which are mentioned as below:
- For the introduction, I suggest the author introduce several key genes that have been reported involved in controlling soybean root traits.
- We have added some key genes regulating soybean root traits in the Introduction Line 78-82.
- Figure 2 is hard to read.
- We have increased the figure 2 size and resolution accordingly.
- In Figure 3, how to understand that the expression value of gene is less than zero?
- We have slightly changed the caption of the figure 3 to make it clearer for knowing the expression greater than or less than zero.
We would again like to thank you for your kind suggestions. Hope we have addressed your comments.
Reviewer 2 Report
Comments and Suggestions for Authors
The topic is relevant and substantively well realized. The main objective of the work was achieved by the authors in a complete manner. The research methods used are adequate and adequately described at each stage of the experiment, which allows for reliable conclusions. The entire experiment was carried out without major objections.
However, the presentation of the results needs improvement, as suggested later in the review. In the discussion, the authors thoroughly analyze their results, supported by examples from the literature, which facilitates understanding of the extensive data set. This research can be a valuable reference for soybean breeders. The conclusions are clearly formulated and point to specific applications of the work's results.
Minor comments and suggestions:
In the tables presenting the results, it is worth standardizing the number of digits, especially in cases where the measurement range is more accurate.
It is recommended to use the abbreviation SD for standard deviation in tables.
Figure 1 needs quality improvement. The scale of the X- and Y-axes for each characteristic should be the same to allow easier comparison of the studied environments and combinations.
In Table 3, it is worth removing the values of 1 on the diagonal if two correlation matrices are presented in one table, as they suggest symmetry, which can be misleading.
Quality improvement is also required for Figure 2.
In Figure S2, the Y-axis in all subdiagonals should be scaled to the same range.
Author Response
Dear Reviewer 2, we thank you for your kind suggestions regarding our manuscript. We have addressed your questions and suggestions in the manuscript and listed them as points down below:
- In the tables presenting the results, it is worth standardizing the number of digits, especially in cases where the measurement range is more accurate.
- We have made the necessary corrections in the table by adding commas in the digits to make it standard.
- It is recommended to use the abbreviation SD for standard deviation in tables.
- We have changed the standard deviation as abbreviation form SD accordingly.
- Figure 1 needs quality improvement. The scale of the X- and Y-axes for each characteristic should be the same to allow easier comparison of the studied environments and combinations.
- We have increased the size of the figure and resolution. Furthermore, we have made the scale of the X-axis and Y-axis same for the same parameters.
- In Table 3, it is worth removing the values of 1 on the diagonal if two correlation matrices are presented in one table, as they suggest symmetry, which can be misleading.
- We have removed the values 1 from the Table 3.
- Quality improvement is also required for Figure 2.
- We have increased the figure size and improved the resolution of the Figure 2.
- In Figure S2, the Y-axis in all subdiagonals should be scaled to the same range.
- We have changed the Y-axis scale and made same for Figure S2.
We would like to thank once again for your valuable comments and hope that the suggestions mentioned are fully addressed.